# IMPROVING FEDERATED LEARNING PERSONALIZATION VIA MODEL AGNOSTIC META LEARNING

## ABSTRACT

Federated Learning (FL) refers to learning a high quality *global model* based on decentralized data storage, without ever copying the raw data. A natural scenario arises with data created on mobile phones by the activity of their users. Given the typical data heterogeneity in such situations, it is natural to ask how can the global model be *personalized* for every such device, individually. In this work, we point out that the setting of Model Agnostic Meta Learning (MAML), where one optimizes for a fast, gradient-based, few-shot adaptation to a heterogeneous distribution of tasks, has a number of similarities with the objective of personalization for FL. We present FL as a natural source of practical applications for MAML algorithms, and make the following observations. 1) The popular FL algorithm, Federated Averaging (McMahan et al., 2017), can be interpreted as a meta learning algorithm. 2) Careful fine-tuning can yield a global model with higher accuracy, which is at the same time easier to personalize. However, solely optimizing for the global model accuracy yields a weaker personalization result. 3) A model trained using a standard datacenter optimization method is much harder to personalize, compared to one trained using Federated Averaging, supporting the first claim. These results raise new questions for FL, MAML, and broader ML research.

## 1 INTRODUCTION

In recent years, the growth of machine learning applications was driven by aggregation of large amounts of data in a datacenter, where a model can be trained using large scale distributed system (Dean et al., 2012; LeCun et al., 2015). Both the research community and general public are becoming increasingly aware that there is a variety of scenarios where this kind of data collection comes with significant risks, mainly related to notions of privacy and trust.

In the presence of user generated data, such as activity on mobile phones, Federated Learning (FL) (McMahan & Ramage, 2017) proposes an alternative approach for training a high quality global model without ever sending raw data to the cloud. The FL system proposed by Google (Bonawitz et al., 2019) selects a sample of available devices and sends them a model to be trained. The devices compute an update to the model based on an optimization procedure with locally available data, and the central system aggregates the updates from different devices. Such iteration is repeated many times until the model has converged. The users' training data does not leave their devices. The basic FL algorithm, Federated Averaging (FedAvg) (McMahan et al., 2017), has been used in production applications, for instance for next word prediction in mobile keyboard (Hard et al., 2018), which shows that Federated Learning can outperform the best model trained in a datacenter. Successful algorithmic extensions to the central idea include training a differential private model (McMahan et al., 2018), compression (Konečný et al., 2016b; Caldas et al., 2018a), secure aggregation (Bonawitz et al., 2017), and a smaller number of always-participating nodes (Yang et al., 2019).

FL applications generally face non-i.i.d and unbalanced data available to devices, which makes it challenging to ensure good performance across different devices with a FL-trained global model. Theoretical guarantees are only available under restrictive assumptions and for convex objectives, cf. Li et al. (2019b). In this work, we are interested in *personalization* methods that adapt the model for data available on each device, individually. We refer to a trained global model as the *initial model*, and the locally adapted model as the *personalized model*. Existing FL personalization work

directly takes a converged initial model and conducts personalization evaluation via gradient descent (Beaufays et al., 2019). However, in this approach, the training and personalization procedures are completely disconnected, which results in potentially suboptimal personalized models.

Meta Learning optimizes the performance after adaptation given few-shot adaptation examples on heterogeneous tasks, and has increasing applications in the context of Supervised Learning and Reinforcement Learning. Model Agnostic Meta Learning (MAML) introduced by Finn et al. (2017) is a solely gradient-based Meta Learning algorithm, which runs in two connected stages; meta-training and meta-testing. Meta-training learns a sensitive initial model which can conduct fast adaptation on a range of tasks, and meta-testing adapts the initial model for a particular task.

Both tasks for MAML, and clients for FL, are heterogeneous. For each task in MAML and client in FL, existing algorithms use a variant of gradient descent locally, and send an overall update to a coordinator to update the global model. If we present the FL training process as meta-training in the MAML language, and the FL personalization via gradient descent as meta-testing, we show in Section 2 that FedAvg (McMahan et al., 2017) and Reptile (Nichol et al., 2018), two popular FL and MAML algorithms, are very similar to each other; see also Khodak et al. (2019).

In order to make FL personalization useful in practice, we propose that the following objectives must *all* be addressed, simultaneously.

    (1) **Improved Personalized Model** – for a large majority of the clients.
    (2) **Solid Initial Model** – some clients have limited or even no data for personalization.
    (3) **Fast Convergence** – reach a high quality model in small number of training rounds.

Typically, the MAML algorithms only focus on objective (1); that was the original motivation in Finn et al. (2017). Existing FL works usually focus on objectives (2) and (3), and take the personalized performance as secondary. This is largely due to the fact that it was not obvious that getting a solid initial model is feasible or practical if devices are available occasionally and with limited resources.

In this work, we study these three objectives jointly, and our main contributions are:

- We point out the connection between two widely used FL and MAML algorithms, and interpret existing FL algorithm in the light of existing MAML algorithms.
- We propose a novel modification of FedAvg, with two stages of training and fine-tuning, for optimizing the three above objectives.
- We empirically demonstrate that FedAvg is already a meta learning algorithm, optimizing for personalized performance, as opposed to quality of the global model. Furthermore, we show that the fine tuning stage enables better and more stable personalized performance.
- We observe that different global models with the same accuracy, can exhibit very different capacity for personalization.
- We highlight that these results challenge the existing objectives in the FL literature, and motivate new problems for the broader Machine Learning research community.

## 2 Interpreting FedAvg as a Meta Learning Algorithm

In this section, we highlight the similarities between the FL and MAML algorithms and interpret FedAvg as a linear combination of a naive baseline and a collection of existing MAML methods.

Algorithm 1 presents a conceptual algorithm with nested structure (left column), of which the MAML meta-training algorithm, Reptile (middle column), and FL-training algorithm, FedAvg (right column), are particular instances. We assume that $L$ is a loss function common to all of the following arguments. In each iteration, a MAML algorithm trains across a random batch of tasks $\{T_i\}$. For each task $T_i$, it conducts an inner-loop update, and aggregates gradients from each sampled task with an outer-loop update. In each training round, FL uses a random selection of clients $\{T_i\}$. For each client $T_i$ and its weight $w_i$, it runs an optimization procedure for a number of epochs over the local data, and sends the update to the server, which aggregates the updates to form a new global model. If we simplify the setting and assume all clients have the same amount of data, causing the weights $w_i$ to be identical, Reptile and FedAvg in fact become the same algorithms. Several other MAML algorithms (Finn et al., 2017; Antoniou et al., 2018), or other non-MAML/FL methods (Zhang et al., 2019), can also be viewed as instances of the conceptual method in left column of Algorithm 1.

**Algorithm 1** Connects FL and MAML (left), Reptile Batch Version(middle), and FedAvg (right).

OuterLoop/Server learning rate $\alpha$
InnerLoop/Client learning rate $\beta$
Initial model parameters $\theta$
**while** not done **do**
   Sample batch of tasks/clients $\{T_i\}$
   **for** Sampled task/client $T_i$ **do**
     **if** FL **then**
       $g_i, w_i = ClientUpdate(\theta, T_i, \beta)$
     **else if** MAML **then**
       $g_i = InnerLoop(\theta, T_i, \beta)$
     **end if**
   **end for**
   **if** FL **then**
     $\theta = ServerUpdate(\theta, \{g_i, w_i\}, \alpha)$
   **else if** MAML **then**
     $\theta = OuterLoop(\theta, \{g_i\}, \alpha)$
   **end if**
**end while**

**Require:** : Reptile Step $K$.
  **function** $InnerLoop(\theta, T_i, \beta)$
    Sample $K$-shot data $D_{i,k}$ from $T_i$.
    $\theta_i = \theta$
    **for** each local step i from 1 to K **do**
      $\theta_i = \theta_i - \beta \nabla_\theta L(\theta_i, D_{i,k})$
    **end for**
    Return $g_i = \theta_i - \theta$
  **end function**
**Require:** : Meta Batch Size $M$.
  **function** $OuterLoop(\theta, \{g_i\}, \alpha)$
    $\theta = \theta + \alpha \frac{1}{M} \sum_{i=1}^{M} g_i$
    Return $\theta$
  **end function**

**Require:** FedAvg Local Epoch $E$.
  **function** $ClientUpdate(\theta, T_i, \beta)$
    Split local dataset into batches $B$
    $\theta_i = \theta$
    **for** each local epoch i from 1 to E **do**
      **for** batch $b \in B$ **do**
        $\theta_i = \theta_i - \beta \nabla_\theta L(\theta_i, b)$
      **end for**
    **end for**
    Return $g_i = \theta_i - \theta$
  **end function**
**Require:** Clients per training round $M$.
  **function** $ServerUpdate(\theta, \{g_i, w_i\}, \alpha)$
    $\theta = \theta + \alpha \sum_{i=1}^{M} w_i g_i / \sum_{i=1}^{M} w_i$
    Return $\theta$
  **end function**

In the following, we rearrange the summands comprising the update formula of FedAvg/Reptile algorithm to reveal the connection with other existing methods – a linear combination of the Federated SGD (FedSGD) (McMahan et al., 2017) and First Order MAML (FOMAML) algorithms (Finn et al., 2017) with different number of steps. For clarity, we assume identical weights $w_i$ in FedAvg.

Consider $T$ participating clients and let $\theta$ be parameters of the relevant model. For each client $i$, define its local loss function as $L_i(\theta)$, and let $g_j^i$ be the gradient computed in $j^{th}$ iteration during a local gradient-based optimization process.

FedSGD was proposed as a naive baseline against which to compare FL algorithms. For each client, it simply takes a single gradient step based on the local data, which is sent back to the server. It is a sensible baseline because it is a variant of what a traditional optimization method would do if we were to collect all the data in a central location, albeit inefficient in the FL setting. That means that FedSGD optimizes the performance of the *initial* model, as is the usual objective in datacenter training. The local update produced by FedSGD, $g_{FedSGD}$, can be written as

$$g_{FedSGD} = \frac{-\beta}{T} \sum_{i=1}^{T} \frac{\partial L_i(\theta)}{\partial \theta} = \frac{1}{T} \sum_{i=1}^{T} g_1^i. \tag{1}$$

Next, we derive the update of FOMAML in similar terms. Assuming client learning rate $\beta$, the personalized model of client $i$, obtained after $K$-step gradient update is $\theta_K^i = U_K^i(\theta) = \theta - \beta \sum_{j=1}^{K} g_j^i = \theta - \beta \sum_{j=1}^{K} \frac{\partial L_i(\theta_j)}{\partial \theta}$. Differentiating the client update formula, we get

$$\frac{\partial U_K^i(\theta)}{\partial \theta} = I - \beta \frac{\partial \sum_{j=1}^{K} g_j^i}{\partial \theta} = I - \beta \sum_{j=1}^{K} \frac{\partial^2 L_i(\theta_j)}{\partial \theta^2}. \tag{2}$$

Directly optimizing the current model for the personalized performance after locally adapting $K$ gradient steps, results in the general MAML update proposed by Finn et al. (2017).

$$g_{MAML} = \frac{\partial L_{MAML}}{\partial \theta} = \frac{1}{T} \sum_{i=1}^{T} \frac{\partial L_i(U_K^i(\theta))}{\partial \theta} = \frac{1}{T} \sum_{i=1}^{T} L_i'(U_K^i(\theta))(I - \beta \sum_{j=1}^{K} \frac{\partial^2 L_i(\theta_j)}{\partial \theta^2}). \tag{3}$$

MAML requires to compute 2nd-order gradients, which can be computationally expensive and creates potentially infeasible memory requirements. To avoid computing the 2nd-order term, FOMAML simply ignores it, resulting in a first-order approximation of the objective (Finn et al., 2017). $FOMAML(K)$ then uses the $(K+1)^{th}$ gradient as the local update, after $K$ gradient steps.

$$g_{FOMAML}(K) = \frac{1}{T} \sum_{i=1}^{T} L_i'(U_K^i(\theta))I = \frac{1}{T} \sum_{i=1}^{T} L_i'(\theta_K^i) = \frac{1}{T} \sum_{i=1}^{T} g_{K+1}^i. \tag{4}$$

Now we have derived the building blocks of the FedAvg. As presented in Algorithm 1, the update of FedAvg, $g_{FedAvg}$, is the average of client updates, which are the sums of local gradient updates. Rearranging the terms presents its interpretation as a linear combination of the above ideas.

$$g_{FedAvg} = \frac{1}{T}\sum_{i=1}^{T}\sum_{j=1}^{K}g_j^i = \frac{1}{T}\sum_{i=1}^{T}g_1^i + \sum_{j=1}^{K-1}\frac{1}{T}\sum_{i=1}^{T}g_{j+1}^i = g_{FedSGD} + \sum_{j=1}^{K-1}g_{FOMAML}(j) \quad (5)$$

Note that interpolating to special case, $g_{FedSGD}$ can be seen as $g_{FOMAML}(0)$ – optimizing the performance after 0 local updates, i.e., the current model. This sheds light onto the existing Federated Averaging algorithm, as the linear combination of algorithms optimizing personalized performance after a range of local updates. Note, however, this does *not* mean that FedAvg optimizes for the linear combination of the objectives of the respective algorithms. Nevertheless, we show in the following section that using $K = 1$ results in a model hard to personalize, and increasing $K$ significantly improves the personalization performance, up until a certain point where the performance of initial model becomes unstable.

## 3 PERSONALIZED FEDAVG

In this section, we present Personalized FedAvg algorithm, which is the result of experimental adaptation of the core FedAvg algorithm to improve the three objectives proposed in the introduction.

We denote $FedAvg(E)$ the Federated Averaging method from Algorithm 1, right, run for $E$ local epochs, weighting the updates proportionally to the amount of data available locally. We denote $Reptile(K)$ the method from Algorithm 1, middle, run in the FL setting for $K$ local steps, irrespective of the amount of data available locally. Based on a variety of experiments we explored, we propose **Personalized FedAvg** in Algorithm 2.

---
**Algorithm 2** Personalized FedAvg
---
1: Run $FedAvg(E)$ with momentum SGD as server optimizer and a relatively larger $E$.
2: Switch to $Reptile(K)$ with Adam as server optimizer to fine-tune the initial model.
3: Conduct personalization with the same client optimizer used during training.
---

In general, FedAvg training with several local epochs ensures reasonably fast convergence in terms of number of communication rounds. Due to complexity of production system, this measure was studied as the proxy for convergence speed of FL algorithms. We find that this method with momentum SGD as the server optimizer already optimizes for the *personalized model* – objective (1) form the introduction – while the *initial model* – objective (2) – is relatively unstable. Based on prior work, the recommendation to address this problem would be to decrease $E$ or the local learning rate, stabilizing initial model at the cost of slowing down convergence (McMahan et al., 2017; Wang & Joshi, 2018) – objective (3).

We propose a fine-tuning stage using $Reptile(K)$ with small $K$ and Adam as the server optimizer, to improve the initial model, while preserving and stabilizing the personalized model. We observed that Adam yields better results than other optimizers, see Table 3 in Appendix A.1, and makes the best personalized performance achievable with broader set of hyperparameters, see Figure 2. The subsequent deployment and personalization is conducted using the same client optimizer as used for training, as we observe that this choice yields the best results for FedAvg/Reptile-trained models.

**Experimental setup.** We use the EMNIST-62 dataset as our primary benchmark (Caldas et al., 2018b). It is the original source of the MNIST dataset, which comes with author id, and noticable variations in style, pen width, size, etc., making it a suitable source for simulated FL experiments. The dataset contains $3400$ users, each with a train/test data split, with a total of $671,585$ train and $77,483$ test images. We choose the first $2,500$ users as the initial training clients, leaving the remaining $900$ clients for evaluation of personalization; these clients are not touched during training. The evaluation metrics are the initial and personalized accuracy, uniformly averaged among all of the FL-personalization clients. This is preferred to a weighted average, as in a production system we care about the future performance on each device, regardless of the amount of data available for personalization. Unless specified otherwise, we use the baseline convolutional model available

in TensorFlow Federated (Ingerman & Ostrowski, 2019)[1], using SGD with learning rate $0.02$ and batch size of $20$ as the client optimizer, and SGD with momentum of $0.9$ and learning rate $1.0$ as the server optimizer. Each experiment was repeated $9$ times with random initialization, and the mean and standard deviation of initial and personalized accuracies are reported. We also use the shakespeare dataset for next-character prediction, split in a similar manner with first $500$ clients used for training and remaining $215$ for evaluation of personalization.

## 3.1 Convergence of FedAvg

In Figure 1, left, we present the convergence of both initial and personalized model during training using the Federated Averaging algorithm. The results correspond to training with $E$ being $2$ and $10$, with visualization of the empirical mean and variance observed in the $9$ replicas of the experiment. Detailed values about performance after $500$ rounds of training, and the number of rounds to reach $80\%$ accuracy, are provided in Table 1. These results provide a number of valuable insights.

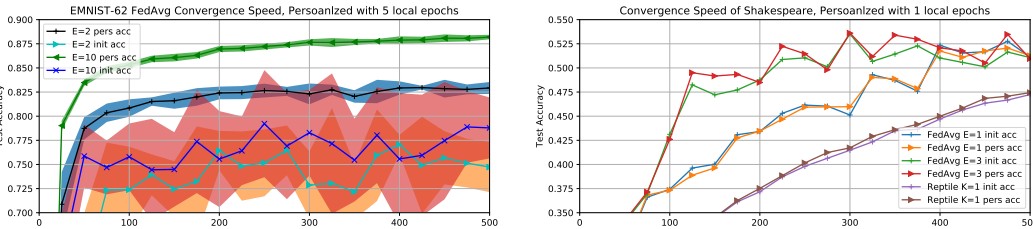

Figure 1: Training Convergence on EMNIST-62 (left) and Shakespeare (right).

First, the personalized accuracy converges significantly higher than the initial accuracy. This clearly validates the EMNIST-62 as an interesting simulated dataset to use to study Federated Learning, with significantly non-i.i.d. data available to each client.

Second, it provides empirical support to the claim made in Section 2, that *Federated Averaging is already a Meta Learning algorithm*. The personalized accuracy not only converges faster and higher, but the results are also of much smaller variance than those of the initial accuracy.

Third, the personalized accuracy after training with $E = 10$ is significantly higher than the personalized accuracy after training with $E = 2$. This is despite the fact that the gap in the initial accuracy between these two variants is somewhat smaller. Moreover, the variance of personalized accuracy is nearly 3-times smaller for training with $E = 10$, compared to $E = 2$, despite the variance of the initial accuracy being smaller for the $E = 2$ case. This supports the insight from Equation 5, that *Federated Averaging with more gradient steps locally should emphasize the personalized accuracy* more, potentially at the cost of initial accuracy.

Finally, this challenges the objectives in the existing literature focusing on the setting of Federated Learning. FedAvg was presented as optimizing the performance of a shared, global model, while in fact it might achieve this only as a necessary partial step towards optimizing the personalized performance. We argue that *in the presence of non-i.i.d. data available to different clients, the objective of Federated Learning should also be the personalized performance*. Consequently, the recommendations that in order to stabilize the convergence, one might decrease the number of local epochs or the local learning rate (McMahan et al., 2017; Wang & Joshi, 2018), are in some scenarios misguided. In this experiment, even though the initial accuracy is very noisy and roughly constant at a suboptimal level, the personalized accuracy keeps increasing.

To evaluate the convergence speed more closely, Table 1 measures the accuracies after $500$ rounds of training and the average number of communication rounds where the initial and personalized accuracy first time reaches $80\%$. While Figure 1 shows results using $5$ clients per round, the table below also shows the same experiment with $20$ clients per round, which in general provides even better and more stable personalized accuracy. The common pattern is that increasing $E$ initially helps, until a certain threshold. From this experiment, $E$ in the range of $5 - 10$ seems to be the best.

We conduct a similar experiment with the Shakespeare data. The result is in Figure 1, right, but does not provide any interesting insights, as the personalized performance shows only a small positive

---

[1]Presented experiments were implemented in TFF, and we will make the code publicly available.

| EMNIST-62 5 clients | Initial Acc | Personalized Acc | Epochs to 0.8 (init/pers) |
|---|---|---|---|
| FedAvg E=2 | 0.7473(0.0260) | 0.8292 (0.0061) | 310.0/63.6 |
| FedAvg E=5 | **0.8028 (0.0512)** | 0.8712 (0.0049) | **111.1**/33.9 |
| FedAvg E=10 | 0.7879(0.0316) | **0.8820 (0.0023)** | 137.5/**30.0** |
| FedAvg E=20 | 0.7430(0.0309) | 0.8782 (0.0021) | 152.5/32.2 |
| EMNIST-62 20 clients | | | |
| FedAvg E=2 | 0.8403 (0.0173) | 0.8957 (0.0011) | 82.5/50.0 |
| FedAvg E=5 | 0.8471 (0.0084) | **0.9057 (0.0017)** | **65.6**/31.25 |
| FedAvg E=10 | **0.8480 (0.0036)** | 0.9032 (0.0017) | 68.7/**25.0** |
| FedAvg E=20 | 0.8391 (0.0081) | 0.8953 (0.0022) | 82.1/46.4 |

Table 1: EMNIST-62 performance for 5 and 20 clients per communication round.

improvement. We conjecture that this is due to the nature of the objective – even though the data is non-i.i.d., the next-character prediction is mostly focused on a local structure of the language in general, and is similar across all users.[2] We thus do not study this problem further in this paper. It is likely that for a next-word prediction task, personalization would make a more significant difference.

## 3.2 BEHAVIOR OF PERSONALIZATION

In this section, we study the ability of models to personalize more closely. In particular, we look at the personalization performance as a function of the number of local epochs spent personalizing, and the effect of both local personalization optimizer and the optimizer used to train the initial model.

In Figure 2, left, we study the performance of three different models, personalized using different local optimizers. The models we test here are: one randomly chosen from the models trained with $E = 10$ in the previous section, with initial accuracy of $74.41\%$. The other two models are the results of fine-tuning that specific model with $Reptile(1)$ and $Reptile(10)$ and Adam as the server optimizer for further $200$ communication rounds, again using $5$ clients per round. For all three models, we show the results of personalization using Adam with default parameters and using SGD with learning rate of $0.02$ and batch size of $100$.

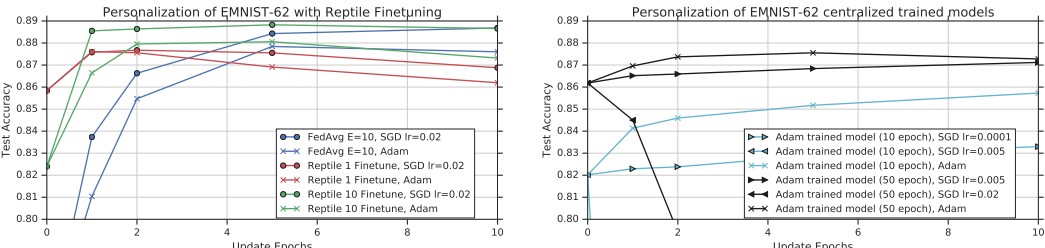

Figure 2: Personalized accuracy as a function of local update epochs. Federated initial models (left), initial models trained by centralizing data and using default Adam optimizer (right).

In all three cases, Adam produces reasonable personalized result, but inferior to the result of SGD. We tried SGD with a range of other learning rates, and in all cases we observed this value to work the best. Note that this is the same local optimizer that was used during training and fine tuning, which is similar to the MAML works, where the same algorithm is user for meta-training and meta-testing.

The effect of fine-tuning is very encouraging. Using $Reptile(10)$, we get a model with better initial accuracy, which also gets to a slightly improved personalized accuracy. Most importantly, it gets to roughly the same performance for a wide range of local personalization epochs. Such property is of immense value for pracical deployment, as only limited tools for model quality validation can be available on the devices where personalization happens. Using $Reptile(1)$ significantly improves the initial accuracy, but the personalized accuracy actually drops! Even though the Equation 5 suggests that this algorithm should not take into account the personalized performance, it is not clear that such result should be intuitively expected – it does not mean it actively supresses personalization, either, and it is only fine tuning of a model already trained for the personalized performance.

---

[2]However, we did not try to replace the default model suggested in TensorFlow Federated.

We note that this can be seen as an analogous observation to those of Nichol et al. (2018), where $Reptile(1)$ yields a model with clearly inferior personalized performance.

Motivated by this somewhat surprising result, we ask the following question: *If the training data were available in a central location, and an initial model was trained using standard optimization algorithms, how would the personalization change?* We train such "centralized initial model" using Adam with default settings, and evaluate the personalization performance of a model snapshot after 10 and 50 training epochs. These two models have similar initial accuracies as the two models fine tuned with $Reptile(10)$ and $Reptile(1)$, respectively. The results are in Figure 2, right.

The main message is that it is significantly harder to personalize these centralized initial models. Using SGD with learning rate of $0.02$ is not good – a significantly smaller learning rate is needed to prevent the models from diverging, and then the personalization improvement is relatively small. In this case, using Adam does provide a better result, but still below the fine tuning performance in Figure 2, left. It is worth noting that the personalized performance of this converged model is similar to that of the model we get after fine tuning with $Reptile(1)$, although using different personalization optimizer. At the moment, we are unable to suggest a sound explanation for this similarity.

At the start of the Section 3, we recommended using Adam as the server optimizer for fine tuning, and here we only presented such results. We did try different optimizers, and found them to yield worse results with higher variance, especially in terms of the initial accuracy. See Appendix A.1 for more details, where we see that all of the optimizers can deliver higher initial accuracy at the cost of slightly lower personalized accuracy.

To strengthen the above observations, we look at the distribution of the initial and personalized accuracies of fine tuned models over multiple experiments. In Figure 3, we look at the same three models as in Figure 2. It is clear that the initial model has a large variance in the initial accuracy, the results are in the range of $12\%$, but the personalized accuracies are only within the range of $1\%$. We chose one model, as indicated by the arrows[3], to be fine tuned with $Reptile(10)$ and $Reptile(1)$. In both cases, fine tuning results in a more consistent results in both the initial and personalized accuracy. Moreover, the best personalized accuracy with $Reptile(1)$ is worse than the worst personalized accuracy with $Reptile(10)$.

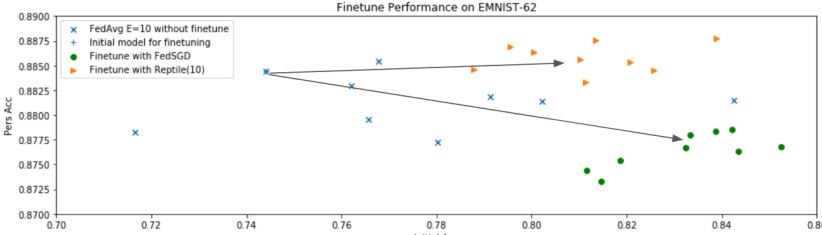

Figure 3: Distribution of initial and personalized accuracies of fine tuned models.

In Appendix A.2, we look at a similar visualization on a per-client basis for a given model. Studying this distribution is of great importance, as in practical deployment, even a small degradation in a user's experience might incur disproportionate cost, relative to the benefit of a comparable improvement in the model quality. We do not study this question deeper in this work, though.

|  | Initial Acc | Personalized Acc |
|---|---|---|
| Reptile(1) Finetuned test clients | 0.8320 (0.0133) | 0.8764 (0.0017) |
| Reptile(1) Finetuned train clients | 0.8577 (0.0019) | 0.8927 (0.0015) |
| Reptile(10) Finetuned test clients | 0.8116 (0.0148) | 0.8858 (0.0014) |
| Reptile(10) Finetuned train clients | 0.8612 (0.0020) | 0.9028(0.0009) |

Table 2: Test performance on clients seen and unseen during FL-training

Finally, we look at the performance of fine tuned models discussed above on both train and test clients. Table 2 shows that if we only looked at the initial accuracy, basic ML principles would

---

[3]The same model was used as the starting point for fine tuning in Figure 2.

suggest the $Reptile(10)$ models are over-fitting, due to larger gap between the train and test clients. However, the personalized accuracy tells a different story - the gap is roughly the same for both model types, and for both train and test clients, $Reptile(10)$ provides significantly better personalized accuracy, suggesting we need a novel way to predict the generalization of personalized models.

## 4 DISCUSSION AND FUTURE WORK

It this work, we argue that in the context of Federated Learning, the accuracy of the global model after personalization should be of much greater interest than it has been. Investigation of the topic reveals close similarities between the fields of Federated Learning and Model Agnostic Meta Learning, and raises new questions for these areas, as well as for the broader Machine Learning community.

**Challenges for Federated Learning.**   Framing papers in the area of Federated Learning (McMahan et al., 2017; Konečný et al., 2016a; Li et al., 2019a), formulate the objective as training of a shared global model, based on a decentralized data storage where each node / client has access to a non-i.i.d sample from the overall distribution. The objective is identical to one the broader ML community would optimize for, had all the data been available in a centralized location.

We argue that in this setting, the primary objective should be the adaptation to the statistical heterogeneity present at different data nodes, and demonstrate that the popular FL algorithm, Federated Averaging, does in fact optimize the personalized performance, and while doing so, also improves the performance of the global model. Experiments we perform demonstrate that the algorithm used to train the model has major influence on its capacity to personalize. Moreover, solely optimizing the accuracy of the global model tends to have negative impact on its capacity to personalize, which further questions the correctness of the commonly presented objectives of Federated Learning.

**Challenges for Model Agnostic Meta Learning.**   The objectives in the Model Agnostic Meta Learning literature are usually only the model performance after adaptation to given task (Finn et al., 2017). In this work, we present the setting of Federated Learning as a good source of practical applications for MAML algorithms. However, to have impact in FL, these methods need to also consider the performance of the initial model,[4] as in practice there will be many clients without data available for personalization. In addition, the connectivity constraints in a production deployment emphasize the importance of fast convergence in terms of number of communication rounds. We suggest these objectives become the subject of MAML works, in addition to the performance after adaptation, and to consider the datasets with a natural user/client structure being established for Federated Learning (Caldas et al., 2018b) as the source of experiments for supervised learning.

**Challenges for broader Machine Learning.**   The empirical evaluation in this work raises a number of questions of relevance to Machine Learning research in general. In particular, Figure 2 clearly shows that models with similar initial accuracy can have very different capacity to personalize to a task of the same type as it was trained on. This observation raises obvious questions for which we currently cannot provide an answer. How does the training algorithm impact personalization ability of the trained model? Is there something we can measure that will predict the adaptability of the model? Is it something we can directly optimize for, potentially leading to novel optimization methods? These questions can relate to a gap highlighted in Table 2. While the common measures could suggest the global model is overfitting the training data, this is not true of the personalized model.

Transfer Learning is another technique for which our result could inspire a novel solution. It is very common for machine learning practitioners to take a trained model from the research community, replace the final layer with a different output class of interest, and retrain for the new task (Oquab et al., 2014). We conjecture that the algorithms proposed in the FL and MAML communities, could yield base models for which this kind of domain adaptation would yield better results.

Finally, we believe that a systematic analysis of optimization algorithms of the inner-outer structure presented in Algorithm 1 could provide novel insights into the connections between optimization and generalization. Apart from the FL and MAML algorithms, Zhang et al. (2019) recently proposed a method that can be interpreted as outer optimizer in the general algorithm, which improves the stability of a variety of existing optimization methods used as the inner optimizer.

---

[4]We recognize that some MAML datasets (Lake et al., 2015) do not admit a good notion of initial accuracy.

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

## A  APPENDIX

This Appendix contains further details referenced from the main body of the paper.

### A.1  FINE TUNING OPTIMIZERS

Table 3 summarizes the attempts at fine tuning the model user in main body with different server optimizers. We see that comparing the same client optimizers, Adam consistently provides better and more stable results in terms of initial accuracy.

| EMNIST-62 Model | Initial Acc | Pers Acc |
|---|---|---|
| FedAvg E=10 w/o finetuning Momentum(lr=1.0, 0.9) | 0.7441 | 0.8842 |
| Finetune with FedSGD (Reptile(1)) Adam | 0.8320 (0.0133) | 0.8764 (0.0017) |
| Finetune with Reptile(5) Adam | 0.8265 (0.0160) | 0.8807 (0.0020) |
| Finetune with Reptile(10) Adam | 0.8116 (0.0148) | 0.8858 (0.0014) |
| Finetune with Reptile(1) SGD(lr=0.1) | 0.8260 (0.0161) | 0.8758 (0.0020) |
| Finetune with Reptile(1) Momentum(lr=0.01, 0.9) | 0.8279 (0.0119) | 0.8745 (0.0014) |
| Finetune with Reptile(5) SGD(lr=0.1) | 0.8148 (0.0280) | 0.8829 (0.0012) |
| Finetune with Reptile(5) Momentum(lr=0.01, 0.9) | 0.8005 (0.0329) | 0.8828 (0.0015) |
| Finetune with Reptile(10) SGD(lr=0.1) | 0.8074 (0.0294) | 0.8855 (0.0012) |
| Finetune with Reptile(10) Momentum(lr=0.01, 0.9) | 0.8140 (0.0246) | 0.8860 (0.0009) |

Table 3: Fine-tuning Result of EMNIST-62 with different server optimizers.

### A.2  PER-CLIENT PERSONALIZATION RESULTS

Figure 4 visualizes the distribution of initial and personalized accuracies on a per-client basis. Each dot represents a random sample of the test clients used for personalization experiments. Studying this distribution is of great importance, as in practical deployment, degrading a user's experience might incur disproportionate cost, compared to the benefit of comparable improvement. Designing methods that robustly identify the clients below the diagonal line and at least revert to the initial model is worth of future investigation.

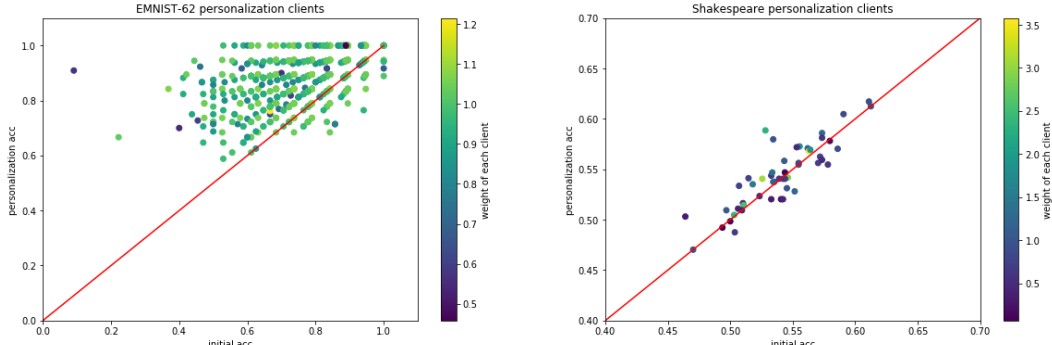

Figure 4: Performance for sample of test clients for EMNIST-62 (left) and Shakespeare (right)

