# OpenReview forum: "Improving Federated Learning Personalization via Model Agnostic Meta Learning"
_ICLR.cc/2020/Conference — Reject_

### Official Review · AnonReviewer1 · 2019-10-23
**Official Blind Review #1**

**Rating:** 1

**Review:**

This paper considers personalization federated learning problem in which the goal is to personalize the global model on a given client/device based on available data on that device/client.  The paper claims not only their proposed method can lead to fast convergence time but also provide a solid initial model per device/client and results in a better-personalized model. To evaluate the performance of their method, EMNIST-62 and Shakespeare data are used.

Even though personalization in federated learning is very interesting and challenging, I am not sure about the contribution of this paper and what is exactly proposed in this paper:

1)  Section 2: this paper shows the relationship between FedAvg and MAML. In my view, the connection is very straight forward and can be shown in a couple of sentences. I might be missing something here, but it is not obvious to me what this paper adds to the connection between MAML and FedAvg.

2)  Personalized FedAvg Section: The same is about section 3. In my view, Algorithm 2 doesn't say anything new rather than to use Adam in local machine and SGD on global models and to optimize for "E" steps. But what if we use other datasets rather than EMNIST-62 and Shakespeare? will these recommendations still hold, i.e. using SGD on server and Adam on the devices? Per section 3 of this paper, Algorithm 2 indeed is the result of the experimental adaptation of the FedAvg algorithm so generalization to other datasets won't be obvious and it is a big question to me.

3) Also, the paper mentioned that this method can work even if there is no local data available on some of the devices/clients. I wasn't able to understand how personalization possible if there is no data to personalize. Wouldn't a device/client just use the global model?

In summary, I find the contribution and novelty of this paper limited and the empirical findings of this paper can't be always applicable to other datasets and scenarios. Plus, I am not convinced this paper shows anything different than FedAvg rather than some recommendations about local and global optimizer selections.

**Experience Assessment:**

I have read many papers in this area.

**Review Assessment: Checking Correctness Of Derivations And Theory:**

I carefully checked the derivations and theory.

**Review Assessment: Checking Correctness Of Experiments:**

I assessed the sensibility of the experiments.

**Review Assessment: Thoroughness In Paper Reading:**

I read the paper thoroughly.

---

> ### Author Response · Authors · 2019-11-11
> **Rebuttal**
>
> We thank the reviewer #1 for their time.
>
> Re: “I am not sure about the contribution of this paper”
> The contribution are the observations that motivate the challenges in Section 4. As formulated in the response to all reviewers, we would like to ask for feedback on these challenges. The contribution is not a novel or superior algorithm - and note we do not claim that in abstract or in the concluding Section.
>
> Re: point 1)
> Even though this connection is not mathematically complex to explain, it has not been formulated, despite FedAvg being proposed before the setting of MAML, and the two fields share many similar characteristics.
> We use this connection to argue and empirically show that what FedAvg was presented to do, is in fact not correct, and is only a side-effect of the actual objective.
>
> Re: point 2)
> See also the response to all reviewers. We do not claim that the algorithm is particularly novel or superior in any sense, but use it to explore new relationships which eventually motivate the challenges in Section 4. We would welcome feedback on what we did, in addition to what we did not do. We agree that experiments on other datasets will be valuable, and see this as orthogonal to the recommendations we are arguing for.
>
> Re: point 3)
> This is what we formulate as the practical requirements in Section 1 - that we need to consider jointly all three of the following objectives - (1) Improved Personalized Model (2) Solid Initial Model and (3) Fast Convergence - with (2) motivated by the fact that many clients will not have data to personalize on. And it is the motivation for what we study in Section 3.2. - start with model with good personalization and improve the initial accuracy.

---

### Official Review · AnonReviewer3 · 2019-10-30
**Official Blind Review #3**

**Rating:** 1

**Review:**

Update: I thank authors for the rebuttal. I agree that direction of exploring personalization in FL is interesting. With a stronger methodological contribution, this could become a good paper.

----------------------------------------------------------------------------------------------------------------
The main contribution of this paper is to notice the connection between Federated Averaging (FedAvg) and Model Agnostic Meta Learning algorithms (MAML). Authors also consider an algorithm that first trains with FedAvg and then continues training using Reptile.

Pros:

Interpretation of FedAvg as a meta-learning algorithm is interesting.

Cons:

Very limited methodological contribution. Proposed algorithm is essentially two existing algorithms applied one after another.

Experiments are not conducted rigorously enough. There are many arbitrary hyperparameter choices which may bias the conclusions made in the paper. Statement "We tried SGD with a range of other learning rates, and in all cases we observed this value to work the best." is alarming suggesting that authors tried a variation of settings observing test data performance and reported a few selected runs. Although "each experiment was repeated 9 times with random initialization", the train/test split of the clients was fixed. Randomizing over client train/test split could help to improve the reliability of the results.

EMNIST-62 is the only dataset analyzed in some detail. This dataset has drastically varying P(y|x) across clients, i.e. some people write zeros as some others write 'o's. This suggests that it is very hard to train a good global model and personalization is necessary. However this doesn't mean that Shakespeare dataset "does not provide any interesting insights". Perhaps, it is indeed more interesting and challenging, demanding more advanced methodology.

In Figure 1, number of communication rounds may be impractical for FL (considering also addition 200 Reptile rounds). On Shakespeare, FedAvg paper reports 54% accuracy achieved in under 50 communication rounds in one of the settings. There are also recent works on improving communication efficiency that were not discussed or studied for personalization quality, e.g. FedProx from "Federated Optimization in Heterogeneous Networks" and PFNM from "Bayesian Nonparametric Federated Learning of Neural Networks".

Questions about Figure 2 experiments:
1. Fine-tuning requires 200 extra epochs over the initially trained model. What's the initial model accuracy when FedAvg is further trained with Adam optimizer for 200 extra communication rounds?
2. The personalized test accuracy with FedAvg and Reptile fine-tuning reaches the same value in 10 update epochs, even when Reptile fine-tuning gets 200 extra initial training epochs. Does Reptile fine-tuning provide additional benefits to the initial model as compared to running FedAvg for more number of epochs?

**Experience Assessment:**

I have published one or two papers in this area.

**Review Assessment: Checking Correctness Of Derivations And Theory:**

I assessed the sensibility of the derivations and theory.

**Review Assessment: Checking Correctness Of Experiments:**

I assessed the sensibility of the experiments.

**Review Assessment: Thoroughness In Paper Reading:**

I read the paper at least twice and used my best judgement in assessing the paper.

---

> ### Author Response · Authors · 2019-11-11
> **Rebuttal**
>
> We thank the reviewer #3 for their time.
> Please also see our shared response to all reviewers - we would like to ask for feedback to the main points presented in this work.
>
> Responding to specific points made:
> Re: Experimental setup / rigor
> Comment on “alarming” choice of learning rate - note we do not make an absolute claim, but a claim relative to the training procedure. The local personalization optimizer (including learning rate) that worked best, was exactly the same as the local optimizer used during training. This is in stark contrast with what we observe for differently trained models in Figure 2, where in one case we had to decrease the learning rate by factor of 200, to prevent the model from diverging. This is of major practical value, as it effectively removes a hyperparameter choice.
> Re: Client train/test split - note that reporting variance would not make sense, as we would not be measuring the same value between such experiments. Moreover, this request is analogous to using different train/test split for any work using (for instance) ImageNet data, which is not aligned with the common practice of the field - so that we would have the same yardstick to compare against, across different publications.
>
> Re: “dataset has drastically varying P(y|x) across clients. (...) it is hard to train a good global model and personalization is necessary”
> Correct. This is the main point, which reflects the practical setting. Nevertheless, almost the entire field has been studying quality of the global model only, following the example of initial paper introducing FedAvg. In Section 4, we argue that the field should shift its focus, and this is one of our main contributions. Also, note that the difference across clients in EMNIST-62 is primarily in P(x) in this dataset, not the conditional.
>
> Re: “this doesn't mean that Shakespeare dataset ...”
> We agree. And we do not make such claim in the submission. We say “even though the data is non-i.i.d., the next-character prediction is mostly focused on a local structure of the language in general, and is similar across all users” which in the language above would be “even though P(x) is different, P(y|x) is similar.”
>
> Re: “FedAvg paper reports 54% accuracy achieved in under 50 communication rounds”
> This does not compare apples and apples. The presentation is not fully reproducible, and is different from the data preprocessing settled on for this dataset in the Leaf and TFF projects (see 1146 clients in FedAvg paper vs. 715 in tff.simulation.datasets.shakespeare due to discarding minor roles) and reports weighted average of accuracies, which we argue is misleading as we care about the future (unknown) performance, and thus report unweighted average.
>
> Re: other recent works:
> We would like to restate the main point of our paper. We do not claim to propose a new algorithm that is superior in some sense. Rather, we challenge the objective studied in existing works, including those you refer to. We agree that studying other methods in the light of objectives we present is valuable, and are trying to argue that the field should do so. As for our main challenge, this is complementary, not contradictory.
>
> Re: Q’s on Fig 2:
> - 75% on average, similar to Momentum, as presented.
> - In Fig. 1, initial accuracy of FedAvg  trained model is stable (in terms of mean/variance across experiments) during rounds 300-500. Further training with the same parameters did not produce a different result.

---

### Official Review · AnonReviewer4 · 2019-11-03
**Official Blind Review #4**

**Rating:** 3

**Review:**

This paper studies the application of techniques from meta-learning (a method
to train a single model which can then be easily adjusted to perform well on
multiple tasks) to federated learning (the task of distributed training of
models on distributed datasets).  The paper notes that standard meta-learning
algorithms are similar to standard federated learning algorithms, and uses
this perspective to produce a merged method and evaluate it empirically.

Pros.
+ The motivation of the paper is clear and indeed these methods seems similar,
 and meta-learning can help with federated learning.

Cons.
- The resulting method appears somewhat underdeveloped; it is simply to run
 some amount of federated learning and then some amount of meta-learning,
 whereas the first parts of the paper led me to believe that a single
 simultaneous merge of the methods is the way to go.  The paper does not
 report any fine-grained evaluation of various such choices, thus I don't know
 why they did that they did, and thus do not find their choices compelling.
- The Reptile method is already presented in the original paper with
 a distributed counterpart, so why not just run that?  I am not convinced that
 some more minor modification of Reptile could not already do well on this
 paper.
- The empirical evaluation is not very extensive, so I am also not convinced
 there, and in particular I need convincing of this type to believe that
 regular reptile is beaten by FedAvg+reptile.

Minor comments.
Page 1, second paragraph, the word "outperform".  I'm not sure what the
performance measure is; in federated learning, we care about many things, for
instance privacy, keeping the work on the distributed clients low, etc.
Page 2, the "three objectives".  I feel meta-learning is doing all three too.
Page 3, Algorithm 1.  I realize space is a concern, but this was hard to read.
Page 4, Algorithm 2.  "relatively larger" is vague.

**Experience Assessment:**

I have published one or two papers in this area.

**Review Assessment: Checking Correctness Of Derivations And Theory:**

I carefully checked the derivations and theory.

**Review Assessment: Checking Correctness Of Experiments:**

I carefully checked the experiments.

**Review Assessment: Thoroughness In Paper Reading:**

I read the paper at least twice and used my best judgement in assessing the paper.

---

> ### Author Response · Authors · 2019-11-11
> **Rebuttal**
>
> We thank the reviewer #4 for their time.
>
> We agree that the method we experiment with, Algorithm 2, is not particularly complex or novel. Note, however, this is not what we present as our main contribution, either. It is not mentioned in the abstract nor the concluding chapter.
>
> Re: “...to believe that regular reptile is beaten by FedAvg+reptile.”
> Related to the above point, we never make such a claim, or that this would even be our objective.
>
> Let us restate the design objectives from Section 1: (1) good initial global model, (2) good personalized model, and (3) fast convergence.
>
> In Section 2, we show that FedAvg and Reptile are essentially the same, with the difference being that FedAvg handles different local data size differently, while this was non-existing concern in the setting in which Reptile was introduced.
>
> Running with FedAvg with large epoch addresses (2) and, mainly (3), but lacks (1). Then switching to a smaller number of steps, independent of the amount of local data (i.e., Reptile) improves (1), without hurting (2) and not needing many additional iterations (3). This contrasts with the experiments in the original Reptile paper, where on the Omniglot task, 40,000 iterations are presented - which would be very expensive in the context of FL.
>
> We also show that (and we don’t find this intuitively expected) Reptile with different number of steps show quite different performance - using K=1 degrades the personalized performance.
>
> In summary, our claim is not that one algorithm “beats” another in a narrow sense, but rather when focusing on the three objectives simultaneously, a combination works better than either of them separately. Fig 2 then shows that models of similar initial accuracy can have very different capacity to personalize, motivating the case for expanding the scope of MAML algorithms, as suggested in the concluding Section 4.
>
> ---
> Re: “the "three objectives".  I feel meta-learning is doing all three too.”
> The problems commonly studied for supervised MAML (random sine wave and Omniglot) do not admit any meaningful notion of initial accuracy - any model is just a random guess. We think this is the main reason why the gap presented in Figure 2 have not been observed before, and that FL applications should become a common part of benchmarks for MAML algorithms.
>
> Re: Distributed Reptile
> We are not sure what you refer to. The paper https://arxiv.org/pdf/1803.02999.pdf has only a single short remark (end of Section 3) on anything related to distributed optimziation.
>
> Please also see our shared response to all reviewers.

---

### Public Comment · ~Stone_Jamess1 · 2019-10-01
**Questions about Equation**

I quite don't understand why the eq(5) is derived.

If you choose the Fedsgd, it means every step you have to update the model and every step the gradient global(g_i) is computed based on the current average global model.

If you use the fedavg, then the local client will compute gradients k steps, but the gradient local(g_i) is computed based on the local model.

If you don't average at every step, then the global model parameter and local model parameter are different, So how can you connect this two together?
Can you explain about it?

---

> ### Author Response · Authors · 2019-10-02
> **Interpretation of Equation (5)**
>
> Thank you for your interest!
>
> You are correct, the subsequent local gradients are computed with respect to different models, and averaged to form a new model only after a number of local steps. This is motivated by the usual high cost of such averaging operation in federated learning. See for instance Figure 1 in (McMahan, 2017) which proposed FedAvg, for empirical visualization that this idea makes sense if you start from the same point, but not if you have two random models. Our eq (5) is thus only a different view on this existing method, providing additional insight into what is it actually optimizing for.

---

> > ### Public Comment · ~Stone_Jamess1 · 2019-10-02
> > **Thanks for your reply**
> >
> > Thanks for your detailed information

---

### Author Response · Authors · 2019-11-11
**Response to all reviewers**

We thank the reviewers for their time, and
- Feel discouraged because none of the reviews provide us with feedback to the main message of our work - Section 4
- Agree that studying datasets other than EMNIST-62 is valuable, but argue that the presented results already challenge existing practices of the field.

We feel our submission may have been read with incorrect expectations. Our argument does not fit into the usual “here is a new algorithm, here is why it is better than the state-of-the-art”. Rather, it presents novel insights, which challenge what is the objective of the state-of-the-art, and argues that different measures should be the object of study of future works in this area.

As such, we hope the work has the potential to become influential, and we seek feedback on these arguments. Much of the reviews we received focus on what we don’t claim to be our contribution.

The most interesting observation, as summarized in Figure 2 - and motivating the main conclusions of our work - is that the same models, trained differently, to a similar initial accuracy, can have very different capacity to personalize to a task of the same type as it was trained on. We are not aware of any observation of this kind in the ML literature. As summarized in the concluding Section 4, we formulate concrete challenges to the main objectives of the existing FL and supervised MAML works, and also motivate questions beyond the areas of FL/MAML.

We also highlight that traditional measures predicting generalization/overfitting are a surprisingly misleading indicator of how well a model can personalize (Table 2).

We would like to ask the reviewers to re-read Section 4 - where we summarize why and how we think this paper can influence the future work of other researchers - and provide feedback on
- Do the presented results support the challenges? If not, why?
- Are the recommendations likely to impact future works in the field? If not, why?
- Do any of these already have an answer? If yes, which?

We would be disappointed to go through the review process without any feedback on these questions.

---

### Decision · Program_Chairs · 2019-12-19

**Decision:**

Reject

**Comment:**

The reviewers have reached consensus that while the paper is interesting, it could use more time.  We urge the authors to continue their investigations.